# Who can I count on: Honor, self-reliance, and family in the United States and Iran

**Peter Wang** [1] *, **Mohammad Atari**[2], **Daphna Oyserman**[1]

**1** Department of Psychology, University of Southern California, Los Angeles, CA, United States of America, **2** Department of Psychological and Brain Sciences, University of Massachusetts Amherst, Amherst, MA, United States of America

* peterpwa@usc.edu

**Data Availability Statement:** All data and analysis syntax are available on openICPSR (https://www.openicpsr.org/openicpsr/project/203903/version/V1/view).

## Abstract

Honor requires that individuals demonstrate their worth in the eyes of others. However, it is unclear how honor and its implications for behavior vary between societies. Here, we explore the tension between competing views about how to make sense of honor–as narrowly defined through self-reliance and self-defense or as broadly defined through strength of character. The former suggests that demonstrating the ability to defend one's self, is a crucial component of honor, while the latter allows the centrality of self-reliance to vary depending on circumstances. To examine these implications, we conducted studies in the U.S., where self-reliance is central to honor, and in Iran, where individual agency must be balanced against the interests of kin. Americans (Studies 1, 2a; $n = 978$) who endorsed honor values tended to ignore governmental COVID-19 measures because they preferred relying on themselves. In contrast, honor-minded Iranians (Study 2b; $n = 201$) adhered to public-health guidelines and did not prefer self-reliance. Moreover, honor-minded Iranians endorsed family-reliance, but did not moralize self-reliance (Study 3; $n = 107$), while honor-minded Americans endorsed family-reliance and moralized self-reliance (Study 3; $n = 120$). Results suggest that local norms may shape how honor is expressed across cultures.

## 1. Introduction

Honor demands that people act to demonstrate their worth in the eyes of others [1]. In psychology, researchers have largely focused on honor and its relation to violence, emphasizing the worth of individuals in deterring aggression through demonstrations of toughness and self-reliance [2–4]. However, honor is also a focus of study in a number of other disciplines, including anthropology, sociology, and political philosophy. These fields have documented the involvement of honor in a wide range of behaviors, some of which have little to do with demonstrations of toughness. They include civic participation in liberal democracies [5], displays of magnanimity [1], and the use of judicial courts to defend one's moral character [6]. These works suggest that behaviors linked to honor may differ across contexts [7]. The implication is that while much of the psychological research on honor centralizes self-reliance, research from other domains implies that self-reliance is only one of many modes by which honor operates.

**Funding:** This work was supported by the Mind and Society Center, University of Southern California, Los Angeles, CA, and the Department of Psychology, University of Southern California, Los Angeles, CA. The funders had no role in study design, data collection and analysis, decision to publish, or preparation of the manuscript.

**Competing interests:** The authors have declared that no competing interests exist.

These competing models make different predictions about when honor is relevant, and which behaviors are honorable.

In the current studies, we explore these models of honor by focusing on self-reliance, a value that psychologists have theorized to be a core aspect of honor [2, 3], and examining the extent to which its role generalizes beyond regional boundaries. We investigate the relation between honor and self-reliance in the U.S. and Iran, two countries with historically distinct perspectives on the self and social roles [8, 9]. If self-reliance is a central component in the logic of honor, then we should see associations between self-reliance and honor in both countries. On the other hand, if the link between honor and self-reliance is conditioned on other cultural values that define individual worth, such as values related to kinship norms or individualistic achievement, we should see that honor predicts different behaviors depending on what is normatively valued in a particular cultural environment.

## 1.1. What is honor?

Honor is a cultural orientation that emphasizes the social worth of individuals and the personal need for others to recognize that worth [1, 2]. As such, honor bridges individual goals to societal values, motivating people to meet the expectations set by others. Honor thereby tends to encourage cooperation within groups and discourage violations of social norms [10, 11].

At this point, perspectives on honor diverge [2, 11–16]. One perspective focuses specifically on themes of retaliation, self-defense, and independence from governing institutions, emphasizing self-reliance and individual action as a means to demonstrate the ability to defend oneself [2, 3, 4]. This view is intuitive to cultural evolutionary accounts of honor, which emphasize the role of honor in deterring aggression in lawless environments [11, 15]. By demonstrating the ability to fend for themselves, individuals signal to others that they are not easy targets for theft and other forms of aggression [15]. This is important in places where authorities cannot reliably enforce cooperative behavior, because individuals must protect themselves without the backing of state institutions [2]. Researchers have documented the presence of self-reliance values among honor cultures across the Mediterranean region [16]. Anthropological evidence indicates that certain honor cultures perceive reliance on law enforcement to address one's grievances as a mark of weakness [17], suggesting a general tension between the pursuit of honor and reliance on government.

Other perspectives take varying views of honor based on specific ethnographies [17], sociological accounts [18, 19], and philosophical worldviews [20]. These perspectives link honor to themes ranging from civic duty [20] to feudal warfare [18]. However, an underlying logic that ties many of these perspectives together is the idea that honor emphasizes strength of character more generally [12, 20]. Since cultural themes can persist beyond their originating conditions [18, 21], honor interacts with new societal conditions, including the existence of centralized institutions, to generate new honor values [13, 14, 18, 22]. This allows self-reliance to be one of many means of demonstrating personal worth [18, 22]. In this perspective, moral values play a major role in how people perceive honor by specifying which qualities are desirable in a person [7]. For example, when duty is a core value, people accord high honor to those who display unquestioning obedience [14, 23]. However, when autonomy is emphasized as an important value, people might see unquestioning obedience as dishonorable, a sign that one lacks the boldness to stand up to authority [5].

Since moral frameworks vary from culture to culture, the attitudes and behaviors that honor entails also vary [14, 17, 19]. While honor in 20th-century Crete entailed open flaunting of laws and disruptive livestock-raiding to prove one's personal abilities [17], military honor in ancient Rome emphasized strict adherence to duty regardless of personal suffering, ambition,

or feelings [14, 23]. The latter example is difficult to reconcile with self-reliance values, as it regards obedience to authority as strength and independent action as weakness. Indeed, the logic of self-reliance in general is not intuitive in military honor, as institutionalized militaries require certain levels of obedience and coordination to operate [18, 23]. In these examples, each kind of honor tracks personal worth in different ways–the first on the bravado and resourcefulness necessary to win personal feuds, and the second on self-control and the loyalty needed to integrate into an authoritative institution. Honor appears to interact dynamically with its moral environment, as local norms determine which behaviors are honorable and which are not. Thus, honor in Turkey includes more attributes of family and close others than honor in the United States, where it is more focused on individual actions and achievement [24]. In Japan, honor evolved in conjunction with transformations in political structure, becoming less violent as the central government stabilized and harmonizing values such as personal duty became prevalent [18]. In parts of the urban U.S., honor can entail particular, bold personal styles that include expensive clothing and prestige accessories as a means of commanding respect from others [25].

We explore the two perspectives (honor-as-self-reliance vs. broader honor-as-strength-of-character) by comparing honor-based sociopolitical attitudes across two countries: the U.S. and Iran. While notions of honor are found in both countries [3, 26, 27], their moral landscapes are very different. This allows us to examine whether self-reliance relates to honor in two countries with distinctive cultural environments. If self-reliance is central to honor, we should see an association between them in both places.

## 1.2. A tale of two countries

In 1893, historian Frederick Jackson Turner put forth his "frontier thesis," asserting that the unforgiving American frontier had shaped the country's character and introduced a rugged individualism that eschewed government interference [8]. In this frontier myth, 19[th]-century pioneers found themselves in a desolate wilderness, without the protections of government, and were forced to rely on their own efforts and wits to survive [28]. It has extended beyond frontier boundaries, becoming a narrative lens by which European-Americans viewed their national character over the decades. Its themes of self-reliance and opposition to government intervention recurred at various point throughout U.S. history, including the Hoover [8] and Reagan [28] administrations. Indeed, adults in the U.S. who rely on the government for economic assistance are often vilified as freeloaders [29].

Meanwhile, late 19[th]-century Persia (modern-day Iran) had been operating heavily on an intense kinship system for centuries, with the extended family forming a central social and economic unit [9, 30]. Persians relied on family for protection and economic support, since household members shared resources [30]. Even potential marriage partners were sought within the limits of one's kinship structure: Iran has historically had high rates of cousin marriage [31]. Despite fundamental changes in government in the 20[th] century, and the separation of extended families due to large-scale industrialization and urbanization, people continue to rely on family as a main source of support [30]. Protection of family continues to be a key concern in Iranian moral values [26], signifying the moral and social centrality of kinship in Iranian society.

Cultural narratives and traditions provide the framework for interpreting behavior and conveying intentions [18, 21, 24, 32]. Since the U.S. and Iran represent cultural contexts with distinct value systems, behaviors that are interpreted as honorable in one context may be irrelevant to honor in the other context.

With the values of rugged individualism, one must tough out their difficulties, relying on personal competence and willpower. Perceived dependence on government support can

signify weakness and a lack of ambition to make it on one's own. Prevailing narratives about self-reliance [8, 33] may lead Americans who value honor to showcase self-reliance to convey strength of character. Within the intensive kinship system of Iran, one draws strength from the family, and must support and coordinate with family members to bring about socioeconomic success. Strict kinship norms may therefore lead Iranians to enact honor by demonstrating fidelity and supporting family-reliance [26, 27]. Of course, these are not the only moral perspectives in the U.S. and Iran, especially given the presence of minority groups with their own distinctive histories, nor are these perspectives always at work, but they may imply general differences in honor values that in turn lead to different behavioral outcomes.

## 1.3. The current studies: COVID-19 as a real-world example of social threat

In the U.S., the COVID-19 pandemic illustrated the importance of self-reliance and highlighted its impact on people's behavior. Many refused to wear face masks [34], refused to socially distance [34], and hoarded consumer goods [35], demonstrating a refusal to follow government recommendations. In the context of the pandemic, a self-reliance perspective would suggest that one should tough it out rather than conform to government intervention (or perhaps, intrusion). Meanwhile, a family-reliance perspective would suggest that one's priority should be to exercise caution and follow trustworthy advice so as not to put the family in danger.

The link between honor and self-reliance became clear on the American side, as willingness to adhere to COVID-19 preventative measures was regionalized, with Southern states (sometimes known as "honor states" [3, 36]) more opposed to COVID-19 public-health strategies [37]. Americans from honor states saw mask-wearing as a sign of weakness and preferred not to be seen with one [38]. Moreover, Americans who endorsed honor-related beliefs expressed greater preference for individual freedom, favored reopening the economy over protecting public health, and opposed social distancing [39]. The link between honor and pandemic self-reliance in less-WEIRD nations, such as Iran, was largely unknown.

In the current studies, we examined the relationships between honor, self-reliance, and public-health attitudes under the threat of COVID-19 in two countries. We examined whether Americans' endorsement of honor values predicted their attitudes toward self-reliance, and whether their attitudes toward self-reliance in turn predicted pandemic public-health attitudes (Studies 1, 2a). Next, we tested the same ideas in Iran: whether honor values in Iran relate to self-reliance and public-health attitudes the same way (Study 2b). In the U.S., we predicted that honor values may lead people to ignore government COVID-19 instructions because they prefer to rely on themselves. Since we originally based our predictions on the honor-as-self-reliance model, we initially made the same predictions for Iran. However, divergence in the results between the countries led us to explore their differences in honor by adopting the broader honor model in Study 3. We tested whether Iranians and Americans who value honor tend to (a) moralize self-reliance outside of the context of public-health adherence and (b) endorse family-reliance.

**H1**: People who endorse honor values show greater preference for self-reliance.

**H2**: People's honor values are indirectly related to their willingness to adhere to public-health guidelines through their preference for self-reliance.

Measures and a timeline of data collection points are available in S1 File. All data and syntax are available on openICPSR (https://www.openicpsr.org/openicpsr/project/203903/version/V1/view).

## 2. Study 1

### 2.1. Methods

We conducted an a priori power analysis in G*Power 3.1.9.2 [40] for a linear regression predicting self-reliance from honor values. This suggested $n$ = 342 to find an $R^2$ increase at 85% power, with an alpha of 0.05 using (for lack of base rate information) small-to-medium effect size guidelines (partial $R^2$ = 0.03, small $R^2$ = 0.01, medium 0.09) [41]. Study 1 began and ended on May 25, 2020.

Studies 1 to 3 all met the ethical guidelines of the University of Southern California Institutional Review Board (IRB) and were approved as exempt for presenting minimal risk to participants. All methods were carried out in accordance with relevant IRB guidelines and regulations. The requirement for consent was waived by the board, but participants were required to agree to the information outlined in a study information sheet provided at the beginning of each study in order to participate.

**2.1.1. Participants.** We aimed for 400 and recruited 402 U.S. participants (final $n$ = 392 after excluding $n$ = 10 who failed the attention check) on Prolific, an online participant recruiting platform, during the last week of May 2020. Inclusion criteria were: U.S. nationality, English fluency, residing in the U.S., and having a Prolific approval rate $\geq$ 90%. Table 1 provides sample descriptions for Studies 1 to 3.

**2.1.2. Measures.** We used a 1 = *Strongly Disagree*, 7 = *Strongly Agree* response option except for political orientation. We detail descriptive statistics, internal consistency coefficients, and the correlations between our scales in Table 2, and measure order and additional measures included for secondary and exploratory analyses in S1 File.

*2.1.2.1. Honor Values Scale (HVS).* Participants responded to the 18-item HVS [42] (α = .86). An example item is "Reputation matters and should be vigorously defended".

*2.1.2.2. Pandemic self-reliance.* We created a 4-item scale (e.g., "During this pandemic, instead of relying on official sources, I should trust myself."; α = .91). We contrasted self-reliance with reliance on government for two reasons: 1) because pandemic-related guidelines came primarily from government sources during this period and 2) because the tension between honor, which is said to flourish in lawless environments [2, 11], and government has been an important theme in prior psychological research. It was therefore important to our aims to specifically investigate self-reliance in relation to government control. In Study 3, we investigated a more general form of self-reliance.

*2.1.2.3. Public health adherence.* We created three items describing actions that adhere to CDC public health guidelines (e.g., "stay at home for as long as possible"). Preliminary analyses (detailed in S1 File) revealed that two items scaled into public health adherence (α = .53). The reliability of our Adherence to Guidelines scale is typical for 2-item scales [43].

*2.1.2.4. Political orientation.* Participants reported their political orientation on social and economic issues (1 = *Very Liberal* to 7 = *Very Conservative*).

### 2.2. Results

**2.2.1. Preliminary analyses.** We examined item skewness and variability and conducted factor analyses (detailed in S1 File). Factors corresponded to our intended constructs: self-reliance and public health adherence. Each of our scales was computed as an average of its items. We explored possible within-U.S. heterogeneity with independent samples t-tests in two ways. First, we compared the South [3] and the rest of the U.S. Second, we compared "red" states that voted for the Republican nominee and "blue" states that voted for the Democratic nominee based in the 2004, 2008, 2012, and 2016 presidential elections. As detailed in S1 File, we

**Table 1. Studies 1 to 4: Sample descriptions.**

| Variable | Study 1 (U.S.) | Study 2a (U.S.) | Study 2b (Iran) | Study 3 (Iran and U.S.) |
|---|---|---|---|---|
| N | 392 | 586 | 201 | 227<br>Iran: 107<br>U.S.: 120 |
| Gender (F = female, M = male) | 42.86% F, 55.36% M, 1.53% other, 0.26% no response | 54.95% F, 43.69% M, 1.37% other | 69.20% F, 30.80% M[a] | Iran: 67.29% F, 23.36% M<br>U.S.: 49.20% F, 50.00% M, 0.80% Other |
| Age in Years | $M = 30.27$, $SD = 10.31$, range 18–68 | $M = 32.27$, $SD = 12.86$, range 18–77 | $M = 34.94$, $SD = 8.32$, range 19–61 | Iran:<br>$M = 21.03$, $SD = 5.52$, range 18–45<br>U.S.:<br>$M = 40.01$, $SD = 13.42$, range 19–79 |
| Race-Ethnicity (multiple selections allowed) | 75.26% Caucasian/White, 15.56% Asian/Asian-American, 8.93% Hispanic/Latino,<br>7.40% Black/African American,<br>2.55% Native American or Alaska Native,<br>0.51% Middle Eastern,<br>0.26% Hawaiian Native or Pacific Islander,<br>1.02% Other | 70.48% Caucasian/White, 17.75% Asian/Asian-American, 8.53% Hispanic/Latino,<br>6.83% Black/African American,<br>1.88% Native American or Alaska Native,<br>1.54% Middle Eastern,<br>0.51% Hawaiian Native or Pacific Islander,<br>0.51% Other | 100% Iranian/Persian, first language Farsi | Iran:<br>100% Iranian/Persian, first language Farsi<br>U.S.:<br>76.70% Caucasian/White, 9.20% Asian/Asian-American, 7.50% Hispanic/Latino,<br>11.70% Black/African American,<br>0.80% Native American or Alaska Native,<br>0.80% Middle Eastern,<br>1.70% Hawaiian Native or Pacific Islander,<br>0.80% Other |
| Political Orientation | $M = 3.03$, $SD = 1.45$; 1 = *Very Liberal*, 7 = *Very Conservative* | $M = 2.98$, $SD = 1.58$; 1 = *Very Liberal*, 7 = *Very Conservative* | 85.07% left-leaning, 14.93% right-leaning[b] | U.S.: $M = 3.23$, $SD = 1.69$; 1 = *Very Liberal*, 7 = *Very Conservative*[c] |

Study 1 after excluding n = 10 who failed the attention check. Study 2a after excluding n = 27 who failed attention check, n = 3 for incomplete scales, and n = 1 was under 18. Study 3 after excluding n = 3 under 18.

[a], [b] binary (male, female; left-leaning, right-leaning).

[c] Study 3 did not ask for political orientation in the Iranian sample.

did not find differences on mean endorsement of honor, self-reliance, and public health adherence.

**2.2.2. H1: Honor is associated with self-reliance.** Americans higher in honor values had greater preference for self-reliance, $\beta = 0.40$, $p < .001$. This relationship remained robust, $\beta = 0.33$, $p < .001$, even after controlling for political orientation, gender (excluding 7 participants who responded "Other" or did not respond on the gender item), and age, detailed in Table 3.

**2.2.3. H2: Indirect effect of honor on public health adherence through self-reliance.** Since the relationships between honor and self-reliance (path *a*; see 2.2.2) and between self-

**Table 2. Study 1: Means, standard deviations, Cronbach's alphas, and correlations for key variables (U.S.).**

| Measure | M (SD) | α | Pearson's *r* Correlations | | |
|---|---|---|---|---|---|
| | | | 1 | 2 | 3 |
| **1. Honor Values** | 4.64 (0.80) | .86 | 1.00 | | |
| **2. Self-Reliance** | 4.67 (1.52) | .91 | .40*** | 1.00 | |
| **3. Adherence to Guidelines** | 5.92 (1.04) | .53 | -.04 | -.20*** | 1.00 |

*$p < .05$

**$p < .01$

***$p < .001$

**Table 3. Studies 1, 2a, and 2b (H1): Results of linear regressions predicting self-reliance.**

| Variable | Statistics | | | | | | | |
|---|---|---|---|---|---|---|---|---|
| | B [95% CI] | SE | T | p | Adj partial $R^2$ | F | df | Adj $R^2$ |
| **Study 1 (U.S.)** | | | | | | | | |
| Honor | 0.33 [0.24, 0.42] | 0.05 | 7.26 | < .001 | 0.12 | 27.90 | 380 | 0.22 |
| Political Orientation | 0.24 [0.14, 0.33] | 0.05 | 5.01 | < .001 | 0.06 | | | |
| Gender (0 = male, 1 = female) | -0.01 [-0.19, 0.18] | 0.09 | -0.06 | .954 | -0.02 | | | |
| Age | 0.13 [0.04, 0.22] | 0.04 | 2.92 | .004 | 0.02 | | | |
| **Study 2a (U.S.)** | | | | | | | | |
| Honor | 0.17 [0.09, 0.26] | 0.04 | 4.14 | < .001 | 0.03 | 22.49 | 572 | 0.13 |
| Political Orientation | 0.27 [0.18, 0.35] | 0.04 | 6.29 | <. 001 | 0.07 | | | |
| Gender (0 = male, 1 = female) | 0.03 [-0.13, 0.19] | 0.08 | 0.36 | .717 | 0.00 | | | |
| Age | 0.02 [-0.05, 0.10] | 0.04 | 0.60 | .552 | -0.00 | | | |
| **Study 2b (Iran)** | | | | | | | | |
| Honor | -0.01 [-0.16, 0.14] | 0.08 | -0.12 | .902 | -0.01 | 0.50 | 196 | -0.01 |
| Political Orientation | 0.29 [-0.13, 0.71] | 0.21 | 1.35 | .178 | 0.00 | | | |
| Sex (0 = male, 1 = female) | -0.02 [-0.33, 0.28] | 0.16 | -0.15 | .879 | -0.00 | | | |
| Age | 0.00 [-0.14, 0.15] | 0.07 | 0.04 | .970 | -0.01 | | | |

[95% CI] = 95% Confidence Interval, Honor = Honor Values Scale Score, Adj = Adjusted

reliance and public health adherence (path *b*), β = -0.22, *p* < .001, were significant, we analyzed a mediation model with paths from honor to self-reliance to public health adherence (see Fig 1 for conceptual model). It revealed a significant indirect effect of honor on public health adherence through self-reliance, β = -0.09, 95% CI [-0.13, -0.04] (*ab*; bootstrapped 5000 times). People who endorsed honor were more likely to prefer self-reliance, which in turn was associated with lower adherence. The indirect effect remained in the same direction but was no longer significant after controlling for political orientation, gender, and age, β = -0.03, 95% CI [-0.06, 0.01] (bootstrapped 5000 times). Since our data were collected cross-sectionally, we conducted an alternate mediation [44] with paths from honor values to public health adherence to self-reliance, revealing no significant indirect effect, β = 0.01, 95% CI [-0.01, 0.03] (bootstrapped 5000 times). Total effects (path *c*), β = -0.04, *p* = .449, and direct effects (path *c'*), β = 0.05, *p* = .385, of honor on public health adherence were not significant.

## 2.3. Discussion

We found support for our H1 and H2 predictions with our sample of American adults. Americans who valued honor were more willing to rely on themselves (H1), and this preference for

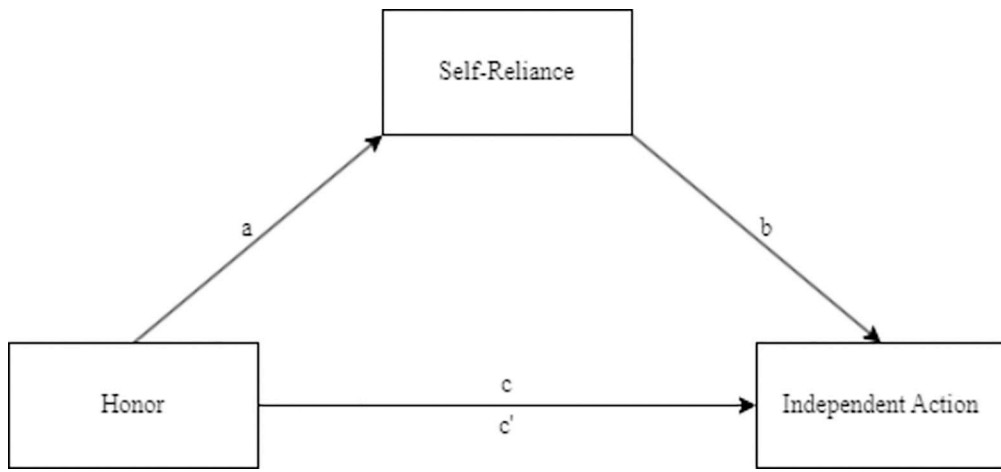

**Fig 1. Conceptual mediation model of the indirect effect of honor on public health adherence through self-reliance.** Path *a* links honor to self-reliance, and *b* links self-reliance to public health adherence. Paths *c* and *c'* describe the total and direct effects, respectively, of honor on public health adherence. The indirect effect of honor on public health adherence through self-reliance is represented as *ab*.

self-reliance explained the association between honor and adherence to public health guidelines (H2). Evidence for H1 was robust to the inclusion of demographic covariates, although H2 results were no longer significant after including the covariates, possibly due to lack of power in controlling for covariates in a mediation model. The alternative model–from honor to adherence to self-reliance, was not significant, increasing our confidence in the predicted direction of the indirect effect. Americans who valued honor were more willing to rely on themselves, and because of this, they were less willing to adhere to CDC-issued public health guidelines.

## 3. Study 2a

In Study 2a, we tested the stability of Study 1 results as the pandemic continued to spread. We wrote the pre-registration for Study 2a on August 19, 2020, before data collection. Note that, however, due to researcher error, our pre-registration was not submitted to AsPredicted.org until September 5, after data collection ended on September 3, but before any data analyses (https://aspredicted.org/blind.php?x=vv7gi8). Moreover, to maintain a focused discussion in this paper, we only present analyses from the pre-registration that are directly relevant to our research question. Remaining analyses are described in S1 File.

### 3.1. Methods

Study 2a data collection began and ended on September 3, 2020.

**3.1.1. Participants.** We aimed for $n$ = 615 U.S. participants based on our a priori G*Power 3.1.9.2 power analysis for our smallest Study 1 effect size (partial adjusted $R^2$ = 0.014), for which we would need $n$ = 614 to find an $R^2$ increase at 85% power. We recruited 617 U.S. adults on Prolific during the first week of September 2020. Inclusion criteria were: U.S. nationality, English fluency, residing in the U.S., Prolific approval rate ≥ 90%, and did not participate in Study 1. Our final sample ($n$ = 586, Table 1, second column for descriptives) excluded people who failed the attention check, skipped key scales, or reported being under 18.

**Table 4. Studies 2a-2b: Variable means, standard deviations, cronbach's alphas, and correlations.**

| Measure | M (SD) | α | Pearson's r Correlations | | |
|---|---|---|---|---|---|
| | | | 1 | 2 | 3 |
| **Study 2a (U.S.)** | | | | | |
| 1. Honor Values Scale | 4.76 (0.86) | .89 | 1.00 | | |
| 2. Self-Reliance | 4.58 (1.68) | .94 | .26*** | 1.00 | |
| 3. Adherence to Guidelines | 6.12 (1.15) | .89 | -.02 | -.24*** | 1.00 |
| **Study 2b (Iran)** | | | | | |
| 1. Honor Values Scale | 5.11 (0.77) | .87 | 1.00 | | |
| 2. Self-Reliance | 2.83 (1.38) | .91 | .03 | 1.00 | |
| 3. Adherence to Guidelines | 6.23 (0.68) | .72 | .18** | -.29*** | 1.00 |

*$p < .05$
**$p < .01$
***$p < .001$

**3.1.2. Measures and procedure.** We used the same response scale of 1 = *Strongly Disagree*, 7 = *Strongly Agree* as in Study 1 and kept our honor ($\alpha$ = .89) and self-reliance scales ($\alpha$ = .94), while adding to our public health adherence items as detailed below ($\alpha$ = .89). We detail descriptive statistics, internal consistency coefficients, and correlations among the study measures in Table 4. Order of measures and additional measures included for exploratory and secondary analyses are detailed in S1 File.

*3.1.2.1. Public health adherence.* The new 3-item scale included our 2 public health adherence items from Study 1, as well as a new item to allow participants to report attitudes toward staying home separate from ability to do so, we added the item: "During the pandemic, I should try to convince my loved ones to stay home as much as possible".

## 3.2. Results

**3.2.1. Preliminary analysis.** Confirmatory factor analyses showed that our items loaded onto our constructs (self-reliance, and public health adherence) as presented in S1 File. Each of our scales was computed as an average of its items.

For exploratory purposes, we used independent samples t-tests to look for within-country variation on honor, self-reliance, and public health adherence. We found greater endorsement of self-reliance in the U.S. South (defined by Cohen and colleagues [3]), as compared with the rest of the U.S, $t(409.35)$ = 3.16, $p$ = .002, Cohen's $d$ = 0.27, though we found no difference between regions that voted Republican as compared to those that voted Democratic in the 2004, 2008, 2012, and 2016 Presidential elections. We present detailed results from these t-tests and explore regional differences further in S1 File.

**3.2.2. H1: Honor is associated with self-reliance.** Supporting H1 and consistent with Study 1, our linear regression, predicting self-reliance from honor revealed that participants who cared more about honor values scored higher in self-reliance, $\beta$ = 0.26, $p < .001$. This relationship remained robust, $\beta$ = 0.17, $p < .001$, even after controlling for political orientation, gender (excluding 8 participants who responded "Other" on the gender item), and age, detailed in Table 3.

**3.2.3. H2: Indirect effect of honor on public health adherence through self-reliance.** Since the relationships between honor and self-reliance (path *a*; see 3.2.2) and between self-reliance and public health adherence (path *b*), $\beta$ = -0.25, $p < .001$, were significant, we analyzed a mediation model with paths from honor to self-reliance to public health adherence (see Fig 1

for conceptual model). It revealed an indirect effect of honor on public health adherence through self-reliance, $\beta$ = -0.07, 95% CI [-0.09, -0.04] (*ab*; bootstrapped 5000 times). People who cared more about honor preferred self-reliance, and self-reliance was associated with lower willingness to adhere to public health guidelines. This was robust to inclusion of political orientation, gender, and age as controls, $\beta$ = -0.02, 95% CI [-0.04, -0.01] (bootstrapped 5000 times). An alternate model with paths from honor to adherence to self-reliance revealed no significant indirect effect of honor, $\beta$ = 0.00, 95% CI [-0.01, 0.02] (bootstrapped 5000 times). Total effects (path *c*), $\beta$ = -0.02, *p* = .640, and direct effects (path *c'*), $\beta$ = 0.05, *p* = .274, of honor on public health adherence were not significant.

## 3.3. Discussion

We predicted that Americans who care more about honor would be less willing to follow government guidelines because they preferred to be self-reliant. The pattern of effects in Study 2a generally replicates those of Study 1 and supports our predictions. Evidence for H1 and H2 were both robust to the inclusion of demographic covariates.

# 4. Study 2b

We examined whether honor is related to self-reliance and public health adherence in Iran, pre-registered https://aspredicted.org/blind.php?x=vb6nv5) on AsPredicted.org. As with Study 2a, to maintain a focused discussion in this paper, we only present analyses from the pre-registration that are directly relevant to our research question. Remaining analyses are described in S1 File.

## 4.1. Methods

Data collection for Study 2b began on November 12, 2020, and ended November 25, 2020.

**4.1.1. Participants.** We conducted a power analysis based on key findings from the Study 2a sample. Based on an a priori power analysis in MedPower [45] to detect the mediating role of self-reliance in the relationship between honor and and another outcome variable detailed in S1 File (specifying standardized coefficients 0.253 for path *a*, 0.532 for path *b*, and 0.118 for path *c'*), aiming for an alpha of 0.05, we needed 158 participants to find an indirect effect at 90% power. We exceeded our goal as we collected data in conjunction with another study, recruiting 201 Iranian participants online through social media advertisements on Telegram from the second to the last week of November 2020. Inclusion criteria were being of Iranian nationality, 18 or older, and speaking Farsi as a first language. No one was excluded from the analyses. See Table 1 for sample description.

**4.1.2. Measures and procedure.** All items were administered through Google Forms. We kept the items that loaded onto our measures from Study 2a (HVS: $\alpha$ = .87, Self-Reliance: $\alpha$ = .91, Adherence: $\alpha$ = .72), making changes for cultural setting. Specifically, we replaced references to the CDC with the relevant Iranian government organization (the Ministry of Health and Medical Education). We asked for political orientation as a binary ("left-leaning", "right-leaning"). We did not ask about race-ethnicity and dropped the attention-check. We detail descriptive statistics, internal consistency coefficients, and correlations among the study measures in Table 4, and measure order and additional measures included for exploratory and secondary analyses in S1 File.

We administered the questionnaires in Farsi. We used Atari and colleagues' Farsi translation of the HVS [26]. We translated the other measures to Farsi and back-translated them to English using the standard back-translation technique [46]. One of the co-authors resolved

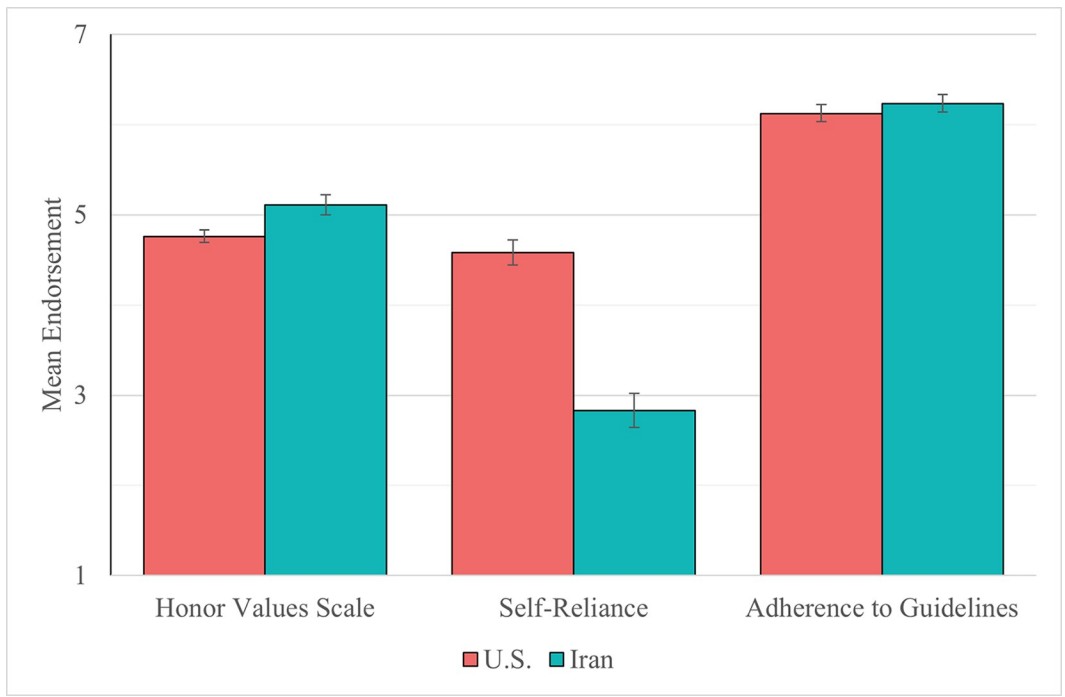

**Fig 2. Mean endorsement of honor, self-reliance, and adherence to guidelines by country.** Error bars indicate 95% confidence intervals.

discrepancies between the two versions, as well as small linguistic and semantic issues in translation. We ensured readability, clarity, and fluency in Farsi.

To address the possibility that our self-reliance items might be understood differently in Iran than in the U.S., we conducted semi-structured cognitive interviewing with a separate sample of 4 Iranians. Participants understood our Farsi self-reliance items as depending on their own beliefs and ideas, matching our intended meaning. We also conducted measurement invariance analyses for our key variables, reported in S1 File.

## 4.2. Results

**4.2.1. Preliminary analysis.** Confirmatory factor analyses showed that our items loaded onto our self-reliance and public health adherence constructs, as presented in S1 File. Each of our scales was averaged.

To examine between-country variations, we investigated differences in construct endorsement between Iran (Study 2b) and the U.S. (Study 2a). We display these comparisons in Fig 2. We found two significant cross-cultural differences: Compared to Americans, Iranians endorsed honor more, Welch-corrected $t(382.31) = 5.42$, $p < .001$, Cohen's $d = 0.42$, and preferred self-reliance less, Welsh-corrected $t(419.61) = 14.68$, $p < .001$, Cohen's $d = 1.09$. Iranians and Americans did not differ in adherence to government guidelines, Welsh-corrected $t(591.18) = -1.57$, $p = .116$, Cohen's $d = 0.10$.

**4.2.2. H1: Honor is associated with self-reliance.** We used linear regression to predict self-reliance from honor values. Honor values and preference for self-reliance were not significantly associated in Iran, $\beta = 0.03$, $p = .692$. As detailed in Table 3, including political orientation and sex as controls did not change the results.

**4.2.3. H2: Indirect effect of honor on public health adherence through self-reliance.** See Fig 1 for the conceptual mediation model. Honor did not predict self-reliance (path *a*; see

4.2.2). Adherence to public-health guidelines positively predicted self-reliance, $\beta$ = -.29, $p <$ .001 (path $b$) while honor positively predicted adherence to guidelines, $\beta$ = .18, $p$ = .010 (path $c$). There was no indirect effect of honor on public health adherence through self-reliance, $\beta$ = -0.01, 95% CI [-0.05, 0.03] ($ab$; bootstrapped 5000 times). The direct effect (path $c'$), $\beta$ = 0.19, $p$ = .005, of honor on public health adherence showed a significant positive relationship. Controlling for political orientation, sex, and age did not make a difference, $\beta$ = -0.00, 95% CI [-0.05, 0.05] (bootstrapped 5000 times).

### 4.3. Discussion

Our results did not support our pre-registered predictions. Among Iranians, endorsing honor values was not associated with preferring self-reliance. Instead, Iranians who endorsed honor values adhered more closely to public-health guidelines. We therefore revised our conceptualization of honor to understand these results. Since Iranians endorsed self-reliance less than Americans, we inferred that Iranians may not moralize self-reliance to the same extent that Americans do and that the differences in results may be due to differences in local norms about self-reliance. We explored these differences further in Study 3.

## 5. Study 3

To further understand the divergence in results from Iran and the U.S., we compared honor-minded Iranians and Americans–whether they tend to treat self-reliance as relevant to morality, and whether they support relying on family. Measuring moralization of self-reliance helps us rule out the possibility that high-honor Iranians in Study 2b still considered self-reliance morally good but were *unable* and therefore unwilling to be self-reliant. Since kinship is important in Iran [26], measuring family-reliance provides insight into how honor might operate: by tapping into core moral themes to determine which behaviors are relevant to honor. While family reliance may not serve as a personal ideal in the same way that self-reliance might, strong kinship ties may normalize relying on family as a socially acceptable approach. Data from the Iranian and American samples were collected at different times, but for simplicity and for the purposes of comparison, we present them together here as one study. Our analysis of the American sample, and comparisons between the Iranian and American data, were pre-registered (https://aspredicted.org/C9F_HW9) on AsPredicted.org.

### 5.1. Methods

Iranian data collection in Study 3 began on July 16, 2022 and ended on July 19, 2022. U.S. data collection began and ended on January 11, 2024.

**5.1.1. Participants.** We recruited a total of 230 participants (110 Iranians, 120 Americans). To determine the American sample size, we conducted a power analysis in G*Power 3.1.9.2 [40], based on initial analysis from the Iranian sample, to find a small-to-moderate correlation (r = .3) with 90% power. Analysis indicated that 113 participants were necessary. We therefore aimed to recruit 120 Americans on Prolific. Inclusion criteria for the American sample were: U.S. nationality, English fluency, residing in the U.S., and Prolific approval rate $\geq$ 90%. Excluding 3 Iranians who reported ages under 18, we included 227 participants in the final analyses. Table 1 provides sample descriptions.

**5.1.2. Measures and procedure.** Items were administered to Iranians through Google Forms in Farsi, and to Americans through Qualtrics in English. We used the same 7-point (1 = *Strongly Disagree* to 7 = *Strongly Agree*) response option set for all Likert-type items. We detail descriptive statistics, internal consistency coefficients, and the correlations between our scales

**Table 5. Study 3: Means, standard deviations, cronbach's alphas, and correlations for key variables.**

| Measure | M (SD) | α | Pearson's r Correlations | | | |
|---|---|---|---|---|---|---|
| | | | 1 | 2 | 3 | 4 |
| **Iran** | | | | | | |
| 1. Honor Values | 4.46 (0.86) | .84 | 1.00 | | | |
| 2. Self-Reliance Moralization | 4.40 (1.83) | .84 | .12 | 1.00 | | |
| 3. Family-Reliance | 4.76 (1.05) | .74 | .35*** | .03 | 1.00 | |
| 4. Self-Control Moralization | 4.20 (1.48) | .94 | .15 | .82*** | .22* | 1.00 |
| **U.S.** | | | | | | |
| 1. Honor Values | 4.85 (0.97) | .91 | 1.00 | | | |
| 2. Self-Reliance Moralization | 4.35 (1.35) | .63 | .42*** | 1.00 | | |
| 3. Family-Reliance | 5.16 (1.00) | .79 | .57*** | .29** | 1.00 | |
| 4. Self-Control Moralization | 3.30 (1.47) | .95 | .39*** | .58*** | .27** | 1.00 |

*$p < .05$

**$p < .01$

***$p < .001$

in Table 5. Due to researcher error, a political orientation item was not included in the Iranian study.

*5.1.2.1. Honor Values Scale (HVS).* We used the same Farsi ($\alpha$ = .84) and English ($\alpha$ = .91) versions of the HVS [26, 42] from Studies 1, 2a, and 2b.

*5.1.2.2. Moralization of self-reliance.* We created a 3-item scale measuring the moralization of self-reliance (e.g., "Relying on others to fulfill my obligations for me."; Iran: $\alpha$ = .84, U.S.: $\alpha$ = .63). Participants were specifically asked to indicate the extent to which they thought each item was morally relevant, not whether it was moral or immoral.

*5.1.2.3. Family-reliance.* We created a 6-item scale measuring willingness to rely on family for support (e.g., "There is nothing wrong with asking your family for support when you are in need."; Iran: $\alpha$ = .74, U.S.: $\alpha$ = .79).

*5.1.2.4. Self-Control Moralization (SCM).* We considered that honor-minded participants may be moralizing behavior generally, not just instances of self-reliance. To account for this possibility, we administered 13 items translated into Farsi from the SCM [47], which was adapted from a measure by Tsukayama and colleagues [48]. Participants were specifically asked to indicate the extent to which they thought self-control failures were morally relevant (e.g., "Being inactive when I have work to do."), not whether they were moral or immoral (Iran: $\alpha$ = .94, U.S.: $\alpha$ = .95). Since the scale instructions were the same as in our moralization of self-reliance scale, we combined the items and presented them as a single scale.

## 5.2. Results

**5.2.1. Preliminary analysis.** As detailed in our Supplementary Materials, confirmatory factor analyses showed that our items loaded onto our self-reliance moralization and family-reliance constructs. Each of our scales was computed as an average of its items.

**5.2.2. Correlating key variables.** Overall, honor scores correlated with endorsement of family-reliance, $r = 0.35$, $p < .001$, and moralization of self-reliance, $r = 0.25$, $p < .001$. Among Iranians, honor scores correlated with endorsement of family-reliance ($r = 0.35$, $p < .001$), but not moralization of self-reliance ($r = 0.12$, $p = .231$) or self-control ($r = 0.15$, $p = .144$). Among Americans, honor scores correlated with endorsement of family-reliance ($r = 0.57$, $p < .001$),

**Table 6. Results of linear regression predicting honor values (Study 3).**

| Variable | Statistics | | | | | | | | |
|---|---|---|---|---|---|---|---|---|---|
| | β [95% CI] | *SE* | *t* | *p* | Adj p $R^2$ | *F* | *df* | Adj $R^2$ |
| **Iran** | | | | | | | | |
| Self-Reliance Moralization | -0.06 [-0.15, 0.26] | 0.10 | 0.54 | .589 | -0.01 | 3.95 | 86 | 0.14 |
| Family-Reliance | 0.42 [0.0.22, 0.62] | 0.10 | 4.18 | < .001 | 0.16 | | | |
| Gender (0 = male, 1 = female) | -0.28 [-0.73, 0.18] | 0.23 | -1.20 | .233 | 0.06 | | | |
| Age | 0.04 [-0.18, 0.26] | 0.11 | 0.34 | .735 | -0.03 | | | |
| Education (1 = high school degree, 2 = Associate's degree and beyond) | -0.12 [-0.66, 0.41] | 0.27 | -0.47 | .643 | -0.01 | | | |
| **U.S.** | | | | | | | | |
| Self-Reliance Moralization | 0.26 [0.11, 0.41] | 0.08 | 3.45 | < .001 | 0.09 | 15.61 | 113 | 0.38 |
| Family-Reliance | 0.44 [0.29, 0.60] | 0.08 | 5.77 | < .001 | 0.22 | | | |
| Gender (0 = male, 1 = female) | 0.01 | 0.15 | 0.10 | .923 | -0.05 | | | |
| Age | 0.16 | 0.07 | 2.24 | .027 | 0.03 | | | |
| Political Orientation | 0.10 | 0.07 | 1.37 | .173 | 0.01 | | | |
| **Overall** | | | | | | | | |
| Self-Reliance Moralization | 0.33 [0.15, 0.51] | 0.09 | 3.58 | < .001 | 0.06 | 15.57 | 201 | 0.33 |
| Family-Reliance | 0.49 [0.32, 0.65] | 0.08 | 5.83 | < .001 | 0.14 | | | |
| Country (0 = U.S., 1 = Iran) | -0.02 [-0.33, 0.30] | 0.16 | -0.11 | .916 | -0.00 | | | |
| Country x Self-Reliance Moralization | -0.28 [-0.52, -0.04] | 0.12 | -2.33 | .021 | 0.02 | | | |
| Country x Family-Reliance | -0.09 [-0.33, 0.15] | 0.12 | -0.71 | .479 | -0.00 | | | |
| Gender (0 = male, 1 = female) | -0.10 [-0.35, 0.14] | 0.12 | -0.84 | .403 | 0.01 | | | |
| Age | 0.17 [0.02, 0.32] | 0.08 | 2.22 | .028 | 0.02 | | | |

[95% CI] = 95% Confidence Interval, Adj = Adjusted, Adj p = Adjusted Partial.

moralization of self-reliance (*r* = 0.42, *p* < .001), and moralization of self-control (*r* = 0.39, *p* < .001).

**5.2.3. Predicting honor from self-reliance moralization and family-reliance.** We used linear regression to predict honor scores from self-reliance moralization and family-reliance (Table 6). To avoid multicollinearity, we did not include moralization of self-control. Iranians who cared more about honor values showed no tendencies to moralize self-reliance, β = 0.11, *p* = .241 but were more accepting toward relying on family, β = 0.35, *p* < .001. Results held even after controlling for gender, age, and education levels (see Table 6), which we measured for exploratory purposes in the Iranian sample. Americans who cared more about honor values tended to moralize self-reliance, β = 0.28, *p* < .001 and support reliance on family, β = 0.49, *p* < .001. Results held even after controlling for gender, age, and political orientation (see Table 6), which we had measured in the U.S. sample but not the Iranian sample. Relevant to

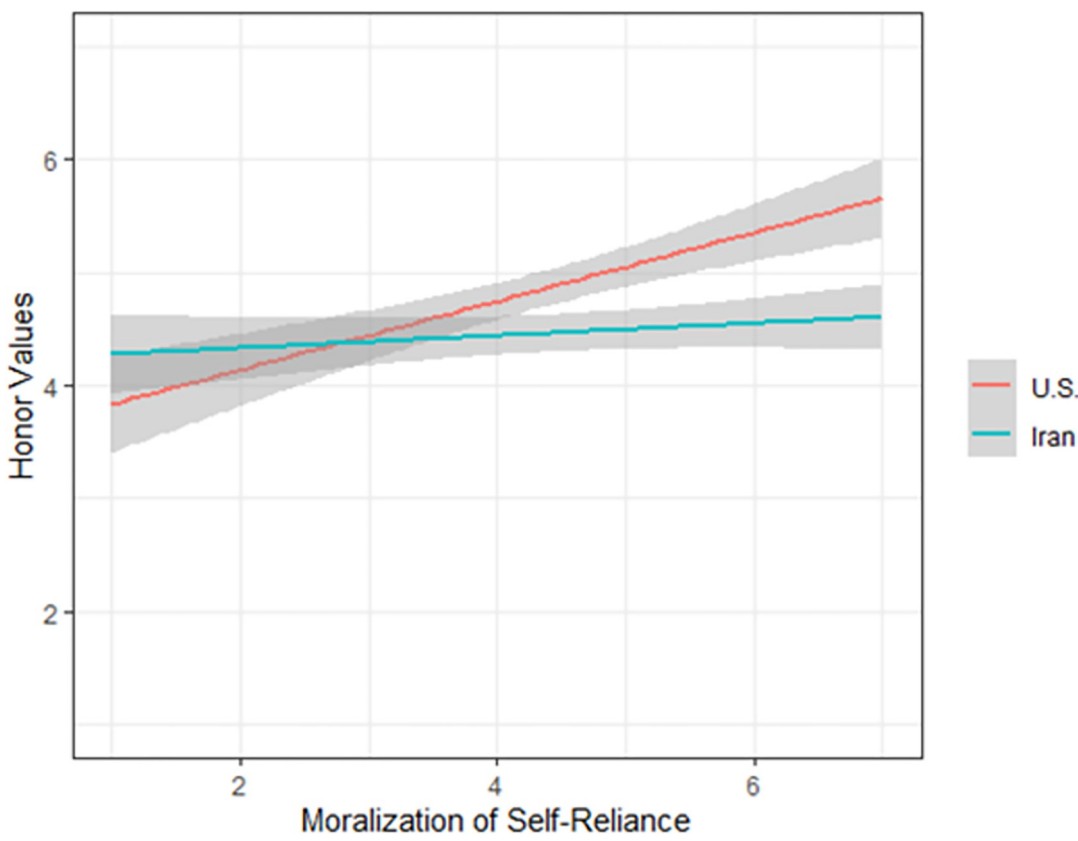

**Fig 3. Interaction of country with moralization of self-reliance (Study 3).** 95% confidence intervals are shaded.

this, moralization of self-reliance correlated positively with endorsement of family-reliance, $r = 0.29$, $p = .001$, suggesting that the two were not mutually exclusive.

We then conducted a linear regression combining the data to predict honor scores from the interactions of country with moralization of self-reliance and endorsement of family-reliance (Table 6). Regression revealed main effects of moralization of self-reliance, $\beta = 0.34$, $p < .001$, and support for family-reliance, $\beta = 0.53$, $p < .001$, such that both were positively associated with honor values. Regression also revealed a main effect of country, such that American participants endorsed honor values more highly than Iranian participants, $\beta = -0.26$, $p = .020$. Importantly, country interacted with self-reliance moralization, $\beta = -0.25$, $p = .032$, such that honor was positively related to the moralization of self-reliance in the U.S., but not in Iran (see Fig 3). There was no interaction of country with support for family-reliance, $\beta = -0.21$, $p = .066$. Results held even after controlling for gender and age, although country no longer predicted honor values (see Table 6).

## 5.3. Discussion

Consistent with our revised concept of honor, moralization of self-reliance was positively associated with honor values in the U.S., but not in Iran. Self-reliance is *not* a key element of Iranian honor; rather, reliance on family is more relevant. Interestingly, support for family-reliance was positively associated with honor values not only in Iran, but also in the U.S., suggesting that solidarity with family may be important to honor in both places. Our family-reliance items measured positive forms of family-reliance, and thus likely did not activate

concerns about freeloading. Supporting this, self-reliance correlated positively with family-reliance, implying that the two constructs were not mutually exclusive.

## 6. General discussion

In Studies 1 to 2b, our predictions were informed by models of honor that focus on independence and self-reliance [2, 4]. However, while we found support for these models in the U.S., we did not find support in Iran. This led us to explore alternate ways of thinking about honor–in particular, we considered that the expression of honor may be determined by other cultural values that indicate what it means to be strong and virtuous [12, 18]. In Study 3, we examined whether people who cared about honor treat self-reliance as a moral problem, as well as their acceptance of relying on family for support. Aligning with American tendencies to treat self-reliance as an indicator of strength of character [8], we found that honor-minded Americans tended to moralize self-reliance. Without such a cultural framework of self-reliance, honor-minded Iranians did not show this tendency.

Interestingly, honor-minded participants from both countries tended to support reliance on family. This suggests that family solidarity may be important in both honor cultures, but more work is needed to understand what that solidarity looks like, and whether they function the same way within the honor cultures (i.e., what information is conveyed when honor-minded Americans vs. Iranians know that someone has a family they can rely on?).

### 6.1. Theoretical implications

More research is certainly needed to test competing conceptions of honor, but our findings provide initial evidence that an honor-as-self-reliance model may not be sufficient for capturing the ways in which honor cultures vary. We may need a more general model of honor that can account for variations between honor cultures while extracting an underlying cultural logic that ties them together. One possible logic is the conception of honor as a demonstration of strength of character, in general. Cultural norms and narratives may determine perceptions of strength, which in turn determines which behaviors are honorable.

This implies that honor does not entail any fixed set of practices or intuitions. Rather, honor follows a logic relevant to agency and power [19, 22]–honor-based actions demonstrate strength of character by expressing qualities aligning with moral values [20]. This is because evaluations of honor are dynamically constructed based on the actor's judgment and intentions [18]. Hence, a "cheap shot" can display weakness and bring dishonor in one setting [49] or display cunning and elicit respect in another setting [22]. While self-reliance and related values may have been tightly linked to honor in the early stages of honor's historical development [11], now it is relevant to honor only to the extent that self-reliance can be used to demonstrate moral prestige. Cultural worldviews that do not designate self-reliance as moral excellence do not link honor to self-reliance. In this way, moral values influence which qualities matter and provide a framework for making sense of one's actions [21], and thus the honor of any particular action depends on (a) the surrounding environment of values and norms and (b) the way in which actors and observers use those values and norms to interpret action. Future research should systematically test this framework to clarify the nature of honor.

### 6.2. Limitations

Limitations of the present research are worth noting. A key limitation is that our mediation models, although consistent with our predictions, are based on cross-sectional data and cannot demonstrate causality. Moreover, reverse mediation testing is often not sufficient for

determining the direction of indirect effects [50]. We cannot assume that the size of our effects would replicate given that cross-sectional data for mediation analysis also leads to biased estimates even in highly controlled conditions [51]. Therefore, we encourage future work to test these models using longitudinal or experimental mediation designs.

Sampling issues present limitations for our studies. While sampling via social media was pragmatically a more accessible way of collecting data from the Iranian population, it tended to recruit younger, female, and more liberal participants, whose views on honor may not represent the broader Iranian population. We also found that our Iranian samples were much younger than our U.S. samples, raising the possibility that differences in honor values were due to differences in age-related experiences. Although controlling for age, political orientation, and gender largely did not change our pattern of results, more work is needed to recruit representative samples and understand how demographic trends, intergenerational differences, and social movements within Iran relate to honor values.

Moreover, we examined moral differences between the U.S. and Iran by measuring values that reflected more dominant narratives and norms in each society. We did not investigate the values of racial-ethnic minorities, who may express honor differently depending on their cultural histories and beliefs. For example, rugged individualism and frontier locality are unrelated for African-Americans, for whom prospects for upward mobility were restricted [8]. Similarly, while Kurds in Iran have faced suppression of their cultural identity and pressure to assimilate to the mainstream culture, they historically incorporated tribal affiliation and feuding practices into their conceptions of honor [52, 53]. As another example, ethnically Arab Iranians tend to have more intensive kinship systems compared with the rest of the country, which in turn could influence their perceptions of honor [54]. Our findings therefore may not necessarily reflect the experiences of specific racial-ethnic minority groups in either country.

Another issue worth noting is that factor analyses, reported in S1 File, showed that the Honor Values Scale did not meet commonly-accepted thresholds for goodness-of-fit. Perhaps part of the problem is that honor values can have substantial heterogeneity not only between cultures, but between individuals within the same culture [18, 55]. Honor measures may need to be broad enough to capture the fundamental logics of honor without narrowing its operationalization to the value systems of specific cultures or subcultures.

More generally, our work was mainly exploratory, pointing to specific ways in which researchers can study honor in the future. To compare different conceptions of honor, researchers will need to structure their studies to systematically test divergent implications corresponding to each model of honor. Differences in constructs such as family-reliance in Iran vs. the U.S. will require further elaboration, since we explored the behavioral implications of self-reliance but did not measure any for family-reliance. Moreover, we saw the COVID-19 pandemic as an important context for understanding the implications of honor and therefore conducted most of our studies during that time. However, the pandemic also made the role of government salient, since individuals had to decide whether or not to trust their governments as sources of pandemic-relevant information. Future studies should examine whether our findings generalize in the absence of a large-scale crisis that enlarges the role of government.

## 7. Conclusion

We began our investigation in the U.S. using an established model of honor that emphasizes the role of self-reliance. We found that honor was associated with self-reliance in the U.S., which was in turn associated with lack of adherence to government guidance during the COVID-19 pandemic. However, results from our studies in Iran suggests that there are cultural boundaries to this model–honor was not associated with self-reliance, and was associated

with more, not less, adherence to public health guidelines. Additional investigation revealed that honor-minded people in the U.S. tend to treat self-reliance as a moral issue, while honor-minded people in Iran do not. Thus, a broader model of honor may be necessary to account for variations in how honor is expressed. We suggest that honor focuses on demonstrations of strength more generally, but how strength is defined depends on culturally-determined moral norms. Our work indicates that models of honor need to account for potential variations in even core themes of honor, but additional work is needed to systematically compare different models of honor.

## Supporting information

**S1 File. Supplemental materials.** Supporting information, containing measures and exploratory and secondary analyses.
(PDF)

**S1 Text. Inclusivity in global research questionnaire.**
(DOCX)

## Author Contributions

**Conceptualization:** Peter Wang, Mohammad Atari, Daphna Oyserman.

**Data curation:** Peter Wang.

**Formal analysis:** Peter Wang, Mohammad Atari.

**Funding acquisition:** Peter Wang, Daphna Oyserman.

**Investigation:** Peter Wang, Mohammad Atari.

**Methodology:** Peter Wang, Mohammad Atari, Daphna Oyserman.

**Project administration:** Peter Wang, Daphna Oyserman.

**Writing – original draft:** Peter Wang.

**Writing – review & editing:** Peter Wang, Mohammad Atari, Daphna Oyserman.

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
