## [Decision Letter · Decision Letter 0]

11 Apr 2024

PONE-D-24-04966Who Can I Count On: Honor, Self-Reliance, and Family in the United States and IranPLOS ONE

Dear Dr. Wang,

Thank you for submitting your manuscript to PLOS ONE. After careful consideration, we feel that it has merit but does not fully meet PLOS ONE’s publication criteria as it currently stands. Therefore, we invite you to submit a revised version of the manuscript that addresses the points raised during the review process.

We look forward to receiving your revised manuscript.

Kind regards,

Amitav Banerjee, M.D.

Academic Editor

PLOS ONE

Journal Requirements:

"This work was supported by the Mind and Society Center, University of Southern California, Los Angeles, CA, and the Department of Psychology, University of Southern California, Los Angeles, CA."

4. Thank you for uploading your study's underlying data set. Unfortunately, the repository you have noted in your Data Availability statement does not qualify as an acceptable data repository according to PLOS's standards.

Additional Editor Comments:

Please revise and respond to the referees' comments.

Reviewers' comments:

Reviewer's Responses to Questions

**Comments to the Author**

1. Is the manuscript technically sound, and do the data support the conclusions?

Reviewer #1: Yes

Reviewer #2: Partly

2. Has the statistical analysis been performed appropriately and rigorously? 

Reviewer #1: Yes

Reviewer #2: Yes

3. Have the authors made all data underlying the findings in their manuscript fully available?

Reviewer #1: Yes

Reviewer #2: Yes

4. Is the manuscript presented in an intelligible fashion and written in standard English?

Reviewer #1: Yes

Reviewer #2: Yes

5. Review Comments to the Author

Reviewer #1: Thank you for giving me the opportunity to review this paper. This paper makes an interesting case about the difference in honor values in Iran and the US. It is timely and important. I believe that honor as a construct in cultural psychology benefits from further scrutiny and this paper presents interesting ideas in this regard.

This paper studied the link between honor and self-reliance in the US and Iran across four studies. In the first three studies, researchers examined the association between honor values, self-reliance, and adherence to Covid health guidelines. They found that self-reliance mediated the link between honor and public health guidelines in the US but not in Iran. In study 4, they found that self-reliance was moralized in the US but not in Iran whereas family reliance was moralized in both cultures. They concluded that honor is construed differently in Iran and the US with the higher emphasis on family reliance in Iran. I find the argument valid and consistent with other cultural works in the Middle East. However, I see several major issues with the writing and operationalization of the paper which undermines the paper’s argument. Below I highlight my suggestions, hoping it would be beneficial to authors to improve their valuable work. Most importantly, I believe running a new study in the post-Covid time with modified self-reliance items and comparable age groups can significantly improve the argument of the paper:

Major issues:

• Writing of the introduction can be improved for clarity. The link between the ideas is not established well and paragraphs lack coherence.

• In their operationalization of self-reliance, authors seem to contrast self-reliance with reliance in the government. This is evidenced by their self-reliance measure: “During this pandemic, instead of relying on official sources, I should trust myself” These two, however, may not be mutually exclusive. One can, for example, have high levels of trust on themselves AND the government. In other words, self-reliance is operationalized to show disobedience to government. Then it is shown to predict lower levels of public health adherence.

• In study 4, the mean age of Iranian participants is 21.03, SD = 5.52, range 18-45 while for American participants it is = 40.01, SD = 13.42, range 19-79. This is a big difference and make the interpretation of the results questionable. These two samples comprise people from different generations, potentially with different values and priorities.

• Study 1-3 study took place during the pandemic. study 4 data collection time in Iran was mid 2022 while it was early 2024 in the US. The results of self-reliance scores, therefore, might be influenced by the salience of disease and Covid health concerns. I suggest replicating the self-reliance findings in the post-covid time as a robustness check.

• In study 4, researchers measured self-reliance moralization but not self-reliance itself. On the other hand, they measured family reliance but not family reliance moralization. What is the reason for this inconsistency? Wouldn’t it be valuable to measure both types for both constructs?

Minor issues:

• The main argument of the paper is not clear in the introduction. My understanding is that self-reliance is argued to be the central theme in the version of honor prevalent in the US South. However, there are other equally important themes in the Iran’s version. If this is the case, it needs to be clearly stated in the introduction.

• The introduction begins with reputation and then transits rapidly to honor without establishing the link between the two: “Maintaining a good reputation is so vital that honor is arguably one of the three features of human culture [5], affecting how people engage with institutions [6,7] and other groups [8]. It’s not clear why and how honor is related to reputation.

• The term “moral conventions” and its relevance to the argument is not explained well. “This perspective allows self-reliance to be one of many means of demonstrating strength, with preference shaped by moral conventions [25,28]. Moral conventions specify the qualities that indicate prestige or superiority, thereby governing how honor relates to those qualities.”

• Study 2 and 3 are similar in their goals and methods. Therefore, I suggest rewriting them as study 2a and 2b.

• On section 5.2.2. there is a mistake in reporting the correlation between honor scores and moralization of self-reliance: “Overall, honor scores correlated with endorsement of family-reliance, r = 0.35, p < .001, and moralization of self-reliance, r = 0.12, p = .231.” The moralization of self-reliance correlation is not significant.

• The political orientation question (left-leaning vs right-leaning) does not seem to be diagnostics in the contemporary Iran for various reasons. Was this item pre-tested to ensure its validity?

• It would be beneficial to report the reliability (e.g. alphas) of the measures for each study in the main text.

Reviewer #2: Dear authors,

Thank you for sending your work to be reviewed in the PLOS ONE. This paper investigates the relationship between honor, self-reliance, and public health attitudes-behaviors under the threat of COVID-19 in two countries (i.e., Iran and the US). Specifically, the paper looked at whether those Americans and Iranians endorsing honor values show a greater preference for self-reliance and if self-reliance, in turn, could mediate the relationship between honor values and adhering to public health guidelines. Additionally, the paper tests whether Iranians and Americans who value honor tend to moralize self-reliance to rule out the possibility that they consider self-reliance good but (for any reason) do not act upon it. The work is theoretically sound and robust, and the arguments in the introduction are solid and persuasive, so I have no comments/concerns about the theoretical aspects of the study. The translations of the Persian versions of measures were exact and sound (I am familiar with Farsi); thus, the authors were successful in measuring the construct they were looking for. In general, I would like to see this paper published in PLOS ONE. However, I have some concerns regarding the methods and implementation of the study, especially the Iranian samples in studies 3 & 4. Please find my comments/concerns below:

* Please note that these comments/concerns are primarily intended to generate a two-way conversation between the reviewer and the author(s). They do not mean that the authors should adhere to and revise every aspect of the review’s concerns. These comments could be addressed by having a two-way conversation between each side.

Major Comments

1. Study 3 did not have attention checks, so no one was excluded from the analysis. Why did the authors decide to drop the attention check in this study? How did they ensure their data quality? How can they be certain that their participants paid enough attention in responding to the surveys, especially when data was recruited from social media, and participants did not receive any compensation?

2. Table S2 supplementary materials: some measures across studies did not show what often considered in psychology good fit indices (CFI > .90, TLI > .90, RMSEA < .06-.08). I appreciate the fact that authors provided results on those measures that a two-factor or even three-factor solutions showed better fits in supplementary, but in some cases, even two-three factor solutions showed relatively weak fits (e.g., Study 4 US. Honor Values, Study 4 Iran Self-Control Moralization). Interpretations, therefore, should be made with caution, and authors should point it out when discussing their limitations.

3. The authors did a great job mentioning the potential limitations of applying this study's findings to other racial-ethnic minorities in the US (e.g., African Americans). While they rightly acknowledge the United States' racial and ethnic diversity, they do not discuss Iran's ethnic diversity. Iran's ethnic composition includes various ethnic groups, such as Persians, Azeris (Turkish), Kurds, Lurs, etc. Therefore, ignoring this diversity and generalizing findings from Persian Iranians (as reported in Table 1, 100% of the sample was Persian and Farsi was their first language) to all Iranians is not a good research practice and should be noticed. We should never assume the other, non-American context is homogeneous, thus avoiding the out-group homogeneity bias.

4. My previous experiences with collecting data from Iranian social media have shown that the sample you usually get is highly educated, liberal (as in study 3), young, has a high rate of people from average to high SES, and is biased toward females (as in study 3 & 4). Therefore, the sample might be different from the actual Iranian demographic. A representative sample may bolster this study's findings, as samples from Iranian social media users tend to be similar to WEIRD populations. Still, it would be highly appreciated if the authors could also mention this as a limitation.

5. Age in Study 4: As shown in Table 1, the mean age difference between the American and Iranian samples is ~19 years old, which is not controlled for in Study 4! It is also possible that differences exist in the political orientations of the two samples (related to my comment above on problems with samples from Iranian social media). As shown in Table 1, the mean political orientation of the American sample is M = 3.23. Although the authors did not report the same construct for the Iranian sample (my first comment is minor concerns), based on the information provided in study 3, we would expect a highly liberal sample of Iranians. I’m unsure if the authors collected other demographics (it seems they have collected education levels from Americans and not Iranians, and it is not clear why). Still, it is highly advised to control for all the possible demographics when comparing samples from two cultures and to report them either in the main manuscript or supplementary materials.

Minor Comments/suggestions

1. Table 1: Information about the political orientation of the Iranian sample needs to be included in Study 4.

2. Instead of presenting the same mediation model three times in three different figures, the authors could provide one figure in which the conceptual mediation model is explained and then report the findings in the manuscript (because they will report them in the manuscript anyway). This would save a lot of space.

3. The same could also be applied to regression tables from study 1 to study 3 (tables 3, 5, and 6).

6. PLOS authors have the option to publish the peer review history of their article (what does this mean?). If published, this will include your full peer review and any attached files.

**Do you want your identity to be public for this peer review?** For information about this choice, including consent withdrawal, please see our Privacy Policy.

Reviewer #1: No

Reviewer #2: **Yes: **Mazyar Bagherian

---

## [Author Response · Author response to Decision Letter 0]

30 May 2024

EDITOR

Comment: 1. Please ensure that your manuscript meets PLOS ONE's style requirements, including those for file naming. The PLOS ONE style templates can be found at 

Response: Thank you for considering our revisions. We have changed the formatting of the paper to meet the requirements of PLOS ONE.

Comment: 2. Please include a complete copy of PLOS’ questionnaire on inclusivity in global research in your revised manuscript. Our policy for research in this area aims to improve transparency in the reporting of research performed outside of researchers’ own country or community. The policy applies to researchers who have travelled to a different country to conduct research, research with Indigenous populations or their lands, and research on cultural artefacts. The questionnaire can also be requested at the journal’s discretion for any other submissions, even if these conditions are not met. Please find more information on the policy and a link to download a blank copy of the questionnaire here: https://journals.plos.org/plosone/s/best-practices-in-research-reporting. Please upload a completed version of your questionnaire as Supporting Information when you resubmit your manuscript.

Response: We have uploaded the inclusivity questionnaire in our submission.

Comment: 3. Thank you for stating the following financial disclosure: 

"This work was supported by the Mind and Society Center, University of Southern California, Los Angeles, CA, and the Department of Psychology, University of Southern California, Los Angeles, CA."

Response: We confirm that this work was supported by the Mind and Society Center, University of Southern California, Los Angeles, CA, and the Department of Psychology, University of Southern California, Los Angeles, CA. The funders had no role in study design, data collection and analysis, decision to publish, or preparation of the manuscript.

Comment: 4. Thank you for uploading your study's underlying data set. Unfortunately, the repository you have noted in your Data Availability statement does not qualify as an acceptable data repository according to PLOS's standards.

Response: We have changed our data repository to openICPSR, made the data publicly available, and changed the link in the text.

Comment: 5. Please include your full ethics statement in the ‘Methods’ section of your manuscript file. In your statement, please include the full name of the IRB or ethics committee who approved or waived your study, as well as whether or not you obtained informed written or verbal consent. If consent was waived for your study, please include this information in your statement as well. 

Response: We have moved our ethics statement to the Methods section on page 9. 

Comment: 6. Please include captions for your Supporting Information files at the end of your manuscript, and update any in-text citations to match accordingly. Please see our Supporting Information guidelines for more information: http://journals.plos.org/plosone/s/supporting-information. 

Response: We have included the Supporting Information section and updated in-text citations. Thank you for helping us to prepare the manuscript for PLOS ONE's guidelines.

REVIEWER 1

Comment: Writing of the introduction can be improved for clarity. The link between the ideas is not established well and paragraphs lack coherence.

Response: We thank Reviewer 1 for their helpful feedback. We have revised the introduction to focus more directly on the research question at hand and avoid discussion of tangential constructs (pages 2-4). We have also elaborated further on certain points in the introduction to make our conceptualization clearer (pages 4, 6, 7, 12).

Comment: In their operationalization of self-reliance, authors seem to contrast self-reliance with reliance in the government. This is evidenced by their self-reliance measure: “During this pandemic, instead of relying on official sources, I should trust myself” These two, however, may not be mutually exclusive. One can, for example, have high levels of trust on themselves AND the government. In other words, self-reliance is operationalized to show disobedience to government. Then it is shown to predict lower levels of public health adherence.

Response: Thank you for this suggestion. We considered self-reliance in opposition to dependence on government partly because this has been an important theme in honor research and because governments were a major source of pandemic-related guidelines. To make our understanding of self-reliance explicit, we elaborated further on self-reliance values in the introduction and methods sections (pages 4, 12). 

Comment: In study 4, the mean age of Iranian participants is 21.03, SD = 5.52, range 18-45 while for American participants it is = 40.01, SD = 13.42, range 19-79. This is a big difference and make the interpretation of the results questionable. These two samples comprise people from different generations, potentially with different values and priorities.

Response: We thank Reviewer 1 for noting this important difference in samples’ age. We added a discussion of the age difference in the Limitations section (page 30) and controlled for age in every study in our new analyses (pages 14, 18, 21, 25, 27).

Comment: Study 1-3 study took place during the pandemic. study 4 data collection time in Iran was mid 2022 while it was early 2024 in the US. The results of self-reliance scores, therefore, might be influenced by the salience of disease and Covid health concerns. I suggest replicating the self-reliance findings in the post-covid time as a robustness check.

Response: We thank Reviewer 1 for this important comment. We agree that it is important to replicate this work in the post-COVID time as a robustness check. In this revision, we opted not to collect new data since it is not feasible for our research team to collect new data from Iran at the moment (there are no easy-to-use platforms like MTurk or Prolific in Iran available to researchers outside the country). However, we found differences in honor values in Studies 1-3 despite the ongoing pandemic and the salience of pandemic-related concerns. We think this suggests that, even when we hold pandemic concerns equal, there are still cross-cultural differences in the kinds of values that honor endorses. Nevertheless, we think that your point makes sense and we have added it to the Limitations section as something that needs to be addressed in future research (page 31).

Comment: In study 4, researchers measured self-reliance moralization but not self-reliance itself. On the other hand, they measured family reliance but not family reliance moralization. What is the reason for this inconsistency? Wouldn’t it be valuable to measure both types for both constructs?

Response: We thank Reviewer 1 for this comment. In hindsight, there are certain measures that would have been useful to include, but we wanted to avoid administering too many measures, especially since the Iranian sample was not compensated for their time. Since we had tested honor-self-reliance associations previously, we opted to focus on self-reliance moralization. Part of our logic is that an honor-based endorsement of self-reliance values would be included within the moralization of self-reliance – honor would need to treat self-reliance as a moral concern before it can treat it as a moral good. We did not measure family-reliance moralization because we did not think of family-reliance as a personal ideal in the same way that self-reliance could be. We thought of family-reliance as something that was normalized by kinship values, but not something that people are told to strive toward. We have elaborated on family-reliance in the paper to make our conception of it more clear (page 22).

Comment: The main argument of the paper is not clear in the introduction. My understanding is that self-reliance is argued to be the central theme in the version of honor prevalent in the US South. However, there are other equally important themes in the Iran’s version. If this is the case, it needs to be clearly stated in the introduction.

Response: Thank you for the suggestion. We agree and have revised the introduction to present our points more clearly (pages 3-4).

Comment: The introduction begins with reputation and then transits rapidly to honor without establishing the link between the two: “Maintaining a good reputation is so vital that honor is arguably one of the three features of human culture [5], affecting how people engage with institutions [6,7] and other groups [8]. It’s not clear why and how honor is related to reputation.

Response: We agree and have removed mention of reputation from the introduction, as we now feel that it is tangential to our main points (pages 3-4).

Comment: The term “moral conventions” and its relevance to the argument is not explained well. “This perspective allows self-reliance to be one of many means of demonstrating strength, with preference shaped by moral conventions [25,28]. Moral conventions specify the qualities that indicate prestige or superiority, thereby governing how honor relates to those qualities.”

Response: We apologize for the lack of clarity. We elaborate on this point further (page 5). We also changed “moral conventions” to “moral values” to make our wording consistent with other parts of the paper and avoid potential confusion.

Comment: Study 2 and 3 are similar in their goals and methods. Therefore, I suggest rewriting them as study 2a and 2b.

Response: We appreciate this suggestion. We have changed the study numbers. We believe this change has made our manuscript more fluent. 

Comment: On section 5.2.2. there is a mistake in reporting the correlation between honor scores and moralization of self-reliance: “Overall, honor scores correlated with endorsement of family-reliance, r = 0.35, p < .001, and moralization of self-reliance, r = 0.12, p = .231.” The moralization of self-reliance correlation is not significant.

Response: We apologize for this mistake. The section should now provide the correct numbers (page 25).

Comment: The political orientation question (left-leaning vs right-leaning) does not seem to be diagnostics in the contemporary Iran for various reasons. Was this item pre-tested to ensure its validity?

Response: Thank you for pointing this out. This item is very similar to what the World Value Survey (WVS) team has been using in Farsi. We did not pre-test this particular item, but we wanted a sense of how our participants chose to identify themselves politically. The fact that most of the participants considered themselves left-leaning was consistent with what we expected from Iranian participants recruited through social media.

Comment: It would be beneficial to report the reliability (e.g. alphas) of the measures for each study in the main text.

Response: Thank you for this suggestion. We have added the alphas in the methods section of each study.

REVIEWER 2

Comment: 1. Study 3 did not have attention checks, so no one was excluded from the analysis. Why did the authors decide to drop the attention check in this study? How did they ensure their data quality? How can they be certain that their participants paid enough attention in responding to the surveys, especially when data was recruited from social media, and participants did not receive any compensation?

Response: Thank you for this comment. An attention check would have been useful; however, our reasoning was that, since participants had no external incentive to complete the study, they would be less likely to select arbitrary responses just to move ahead in the study. 

Comment: 2. Table S2 supplementary materials: some measures across studies did not show what often considered in psychology good fit indices (CFI > .90, TLI > .90, RMSEA < .06-.08). I appreciate the fact that authors provided results on those measures that a two-factor or even three-factor solutions showed better fits in supplementary, but in some cases, even two-three factor solutions showed relatively weak fits (e.g., Study 4 US. Honor Values, Study 4 Iran Self-Control Moralization). Interpretations, therefore, should be made with caution, and authors should point it out when discussing their limitations.

Response: We thank Reviewer 2 for this comment. This is a good point, and we have included this in our limitations (page 31). We suspect part of the issue is that honor scales tend to draw on particular values, and there is likely heterogeneity in how people organize these values.

Comment: 3. The authors did a great job mentioning the potential limitations of applying this study's findings to other racial-ethnic minorities in the US (e.g., African Americans). While they rightly acknowledge the United States' racial and ethnic diversity, they do not discuss Iran's ethnic diversity. Iran's ethnic composition includes various ethnic groups, such as Persians, Azeris (Turkish), Kurds, Lurs, etc. Therefore, ignoring this diversity and generalizing findings from Persian Iranians (as reported in Table 1, 100% of the sample was Persian and Farsi was their first language) to all Iranians is not a good research practice and should be noticed. We should never assume the other, non-American context is homogeneous, thus avoiding the out-group homogeneity bias.

Response: We are glad that the reviewer thinks we have done a great job mentioning the limitations of our cross-cultural research. We revised the section to include heterogeneity in Iran as well, with Kurds as an example of a minority group whose cultural history did not follow the same trajectory as the Persian majority (pages 30-31).

Comment: 4. My previous experiences with collecting data from Iranian social media have shown that the sample you usually get is highly educated, liberal (as in study 3), young, has a high rate of people from average to high SES, and is biased toward females (as in study 3 & 4). Therefore, the sample might be different from the actual Iranian demographic. A representative sample may bolster this study's findings, as samples from Iranian social media users tend to be similar to WEIRD populations. Still, it would be highly appreciated if the authors could also mention this as a limitation.

Response: We thank the reviewer for bringing this point up. We fully agree. We have added this as a limitation (page 30).

Comment: 5. Age in Study 4: As shown in Table 1, the mean age difference between the American and Iranian samples is ~19 years old, which is not controlled for in Study 4! It is also possible that differences exist in the political orientations of the two samples (related to my comment above on problems with samples from Iranian social media). As shown in Table 1, the mean political orientation of the American sample is M = 3.23. Although the authors did not report the same construct for the Iranian sample (my first comment is minor concerns), based on the information provided in study 3, we would expect a highly liberal sample of Iranians. I’m unsure if the authors collected other demographics (it seems they have collected education levels from Americans and not Iranians, and it

---

## [Decision Letter · Decision Letter 1]

18 Jun 2024

Who Can I Count On: Honor, Self-Reliance, and Family in the United States and Iran

PONE-D-24-04966R1

Dear Dr.Wamg

We’re pleased to inform you that your manuscript has been judged scientifically suitable for publication and will be formally accepted for publication once it meets all outstanding technical requirements.

Kind regards,

Amitav Banerjee, M.D.

Academic Editor

PLOS ONE

---

## [Editor Report · Acceptance letter]

18 Jul 2024

PONE-D-24-04966R1 

PLOS ONE

Dear Dr. Wang, 

I'm pleased to inform you that your manuscript has been deemed suitable for publication in PLOS ONE. Congratulations! Your manuscript is now being handed over to our production team.

Kind regards, 

on behalf of

Dr. Amitav Banerjee 

Academic Editor

PLOS ONE